# SEARCH AND RETRIEVAL IN SEMANTIC-STRUCTURAL REPRESENTATIONS OF NOVEL MALWARE

## ABSTRACT

In this study we present a novel representation for binary programs, which captures semantic similarity and structural properties. Our representation is composed in a bottom-up approach and enables new methods of analysis. We show that we can perform search and retrieval of binary executable programs based on similarity of behavioral properties, with an adjustable level of feature resolution. We begin by extracting data dependency graphs (DDG), which are representative of both program structure and operational semantics. We then encode each program as a set of graph hashes representing isomorphic uniqueness, a method we have labeled DDG Fingerprinting. Next, we use k-Nearest Neighbors to search in a metric space constructed from examples. This approach allows us to perform a quantitative analysis of patterns of program operation. By evaluating similarity of behavior we are able to recognize patterns in novel malware with functionality not previously identified. We present experimental results from search based on program semantics and structural properties in a dataset of binary executables with features extracted using our method of representation. We show that the associated metric space allows an adjustable level of resolution. Resolution of the features may be decreased for breadth of search and retrieval, or as the search space is reduced, the resolution may be increased for accuracy and fine-grained analysis of malware behavior.

## 1 INTRODUCTION

In this section we briefly review work related to malware analysis and machine learning, and discuss graph features.

### 1.1 BACKGROUND AND RELATED WORK

Machine learning techniques have been applied in many contexts to accurately classify benign and malicious programs based on various features. Various classification methods have been used, such as deep neural networks and support vector machines. These methods of classification are trained on labeled datasets. An ongoing goal and difficulty in the application of machine learning to malware detection has been to accurately represent the semantic properties of programs. There are several existing approaches to analyze a program based on its behavior, including static and dynamic analysis, or execution traces. Useful features for classification can be extracted at multiple points in the architectural hierarchy. Some of these features are assembly instructions; n-gram sequences of instructions and system calls, and program metadata; patterns of bytecode or hex representations; as well as graphs, n-grams, and sequences of system API calls. Among the most common representations are term frequency (**tf-idf**) features, and data flow of functions in high level languages. However, accurate classification of malicious programs based on their behavior and operational semantics presents several obstacles. Classification is highly dependent on features used in training and increasing resolution beyond class labels poses a challenge. An in depth discussion of feature resolution increase is presented in the Results section (section 3.2) Souri & Hosseini (2018), Deldar & Abadi (2023), Zhou et al. (2019), Wang et al. (2020), Park et al. (2012).

Programs of different classes may have highly similar functionality. Since malware attempts to disguise its operation, behavior may overlap between classes. Obfuscation presents a challenge in

using behavior to distinguish between classes, and increases in resolution are required to provide more interpretable results Souri & Hosseini (2018), Rawashdeh et al. (2021), Kebede et al. (2017), Djaneye-Boundjou et al. (2019), Chandrasekaran et al. (2020).

Several studies have focused on control flow graph datasets and their use in classification. A number of studies explored the use of static features of file metadata. Decision trees for the classification of Windows PE files have been effective for classification. Subsequent studies have used ensemble methods, random forests, and support vector machines, with features extracted from file headers in Trojan malware. In previous studies we have explored cluster analysis and latent semantic analysis of malicious binaries using term frequency representations Bruschi et al. (2006), Cesare & Xiang (2010b), Cesare et al. (2013), Cesare & Xiang (2010a), Shafiq et al. (2009), Siddiqui et al. (2008), Witten et al. (1999), Musgrave et al. (2020), Musgrave et al. (2022).

Hashing of features has been performed in several studies applying machine learning to malware analysis. The focus of the hashing is often to capture the semantics of a function in a high level language such as C or Java. Jang et al. successfully used a hash function on features of binary n-gram sequences to represent malicious programs. These were compared for similarity by their Jaccard index. The focus of their work was an approach from unsupervised learning, and an analysis of the clusters of the hashes obtained. They used a co-clustering approach to demonstrate feature correlation, and also implemented $k$-Nearest Neighbors classification, with precision and recall above 90%. Their features focused primarily on binary stings, but can be extended by the development of a custom hashing function. Liang et al. applied partial order preserving hashing via Gödel hashes to obtain an increase in algorithm performance on existing benchmarks for program flow analysis Jang et al. (2011), Liang et al. (2014).

Several studies have focused on function abstraction semantics through decompilation. LeDoux et al. represented a program as a graph of function abstractions obtained from reverse engineering and used semantic hashing as a measurement of similarity. However, this study did not take a bottom up approach, and basic block features were specifically not considered. There may be many equivalent programs for a given malware binary, and whether semantic function abstractions in a high level language are correlated to lower level binary representations is an open question. In a similar manner, Alrabaee et al. have used a $tf - idf$ representation with Hidden Markov Models and graph kernels to obtain a graph of semantic function abstractions for a program. This was accomplished by constructing a Bayesian network for each of the features collected LeDoux et al. (2013), Alrabaee et al. (2018).

More recently Large Language Models have demonstrated a significant step forward in representing executable programs' semantics. However, the ability of deep learning models to capture program semantics is not matched by increased explainability of the obtained models. Our primary focus in the current work is the problem of gaining greater increases in accuracy and insight into program semantics Xu et al. (2022).

## 1.2 MOTIVATION

In an adversarial environment, malicious programs may be encountered which have not been seen previously and which contain vulnerabilities that are unknown. The problem of classifying operational semantics of previously unseen malware with unknown vulnerabilities is an active area of research.

A classifier's ability to generalize over unseen data is critical for its successful application to novel malware identification. This requires the ability to generalize to abstractions above syntax, to identify patterns and their underlying generative processes, but also a fine grained resolution of interpretable features. Features representing operational semantics are a necessary step in the classification process of zero-day vulnerabilities.

In this study we intend to show that search and retrieval of programs based on semantics can be successfully performed on unseen malware samples without prior training. We demonstrate a method of representing program operational semantics through the construction of features. By representing a program via semantic features, classification can be focused on operational semantics with increased ability to interpret results. By using these features, specific characteristics of patterns are able to be

identified. Programs are able to be compared in relative terms of their operation, and questions of functional class overlap between samples are able to be answered.

We intend to demonstrate that graphs of data dependencies between operands are correlated to both program structure and operational semantics. This representation can be used as a basis for further classification. We have called this method DDG Fingerprinting. The construction of a metric space for this representation allows for search and the evaluation of similarity between programs at a fine grained level. The resolution of the search space is able to be adjusted, and refinement of the search based on specificity leads to more accurate results. Comparison across platform architectures is possible to perform with our approach, although it leads to an increase in the search space and a decrease in resolution. We intend to pursue this in future studies, but include it briefly in this study as a demonstration of increased robustness of the proposed feature representation. Finally, the proposed representation produced by means of DDG Fingerprinting is more explainable than existing approaches and can be easily interpreted by a data analyst.

### 1.3 OUTLINE

Section 2 is a description of the method of data collection, feature construction, the construction of the metric space, and search procedure. Section 3 presents the experimental results. Section 4 contains a discussion of results and conclusions. Higher resolution images of the metric space are presented in the Appendix.

## 2 METHODOLOGY

The following section is a description of the data collection process and methodology followed for experiments.

### 2.1 DATA COLLECTION

We begin by collecting a dataset of benign samples. Each benign sample is deconstructed into its functional components. From this set of functional components we build a library of examples of operational semantic behavior. Benign program binaries for Windows were taken from the $System32$ directory. This system directory contains benign programs that perform standard operating system functions on the Windows platform. For the library of benign functionality we use a set of 500 programs taken from the Windows system directories.

Malicious samples were taken from the public malware repository $theZoo$ for Windows and Linux. The malicious class exemplars we have chosen are the $Win32.APT28.SekoiaRootkit$ and the $ZeusGameover\_Feb2014$ Trojan malware. We briefly include a cross-platform example for comparison of similarity between platforms, and these samples were taken from the $/usr/bin$ directory on Linux for benign samples ytisf.

We have performed our analysis on artifacts of live malware binaries. We selected specific class exemplars for malicious programs from domain knowledge, and evaluated these samples in relative terms to a set of known benign functionality.

While elements in the dataset are labeled as malicious and benign, this class label represents the binary as a whole, and not specific functionality. Determining which functional components are present in a given binary is a critical question, as obfuscation of functionality is the primary goal of a malicious actor.

### 2.1.1 REVERSE ENGINEERING

Given a binary artifact, we perform reverse engineering on the binary to obtain its x86/64 assembly representation. This was done with the GNU $objdump$ utility. The result of this step is a single document containing the equivalent assembly representation of the program Intel-Corp, Free-Software-Foundation.

```
mov      ecx , rbp − 44
mov      eax , ecx
and      eax , 400
or       eax , 140
or       ecx , 1
cmp      rip + 170, 0
cmovne   ecx , eax
mov      rbp − 44, ecx
mov      rip + 180, 0
jmp      0x100000000
```

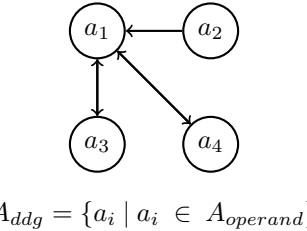

$$A_{ddg} = \{a_i \mid a_i \ \in \ A_{operand}\}$$

Figure 1: A single basic block of assembly instructions and a corresponding directed graph representing data dependencies. The data dependency graph shown is constructed from dependencies between data movement instructions. Nodes in the graph represent data operands. An instruction that moves data between two operands creates a data dependency. We represent this relationship by adding an edge between two operands. Four data operands are present in the basic block. Two register direct addresses in $ecx$ and $eax$, and two register offset addresses in $rbp$ and $rip$. The $ecx$ register has the highest number of dependencies as well as the highest degree in the graph, with three edges from $eax$, $rbp$ and $rip$.

## 2.2 DDG FINGERPRINTING

In this section we present a novel representation for features of malicious programs. This representation is based on hashes of data dependency graphs, which are directly tied to both the structure and operational semantics of a program.

### 2.2.1 SEGMENTATION

We have segmented the program document as a whole into basic block segments. This allows us to increase the feature resolution to be more fine grained, specifically at the level of basic block resolution rather than the level of the program as a whole. Each segment is a basic block of contiguous instructions that are separated by a jump instruction ($jmp$), or other control transition instruction. We split the document into segments based on these jump instructions Musgrave et al. (2022).

### 2.2.2 DATA DEPENDENCY GRAPH EXTRACTION

Any operand using the result of a previous instruction creates a data dependency. We represent these data dependency relationships between the operands within a program segment in the form of an undirected graph corresponding to the data-dependency graph (DDG). The graph's nodes are operands from data movement operations in the segment, and an edge is placed between nodes representing the two operands in a $mov$ instruction. Figure 1 shows a code block and its dependency graph.

In comparison to $tf-idf$ representations of programs, where a single term ($mov$) captures a majority of variance in the term frequency distribution, we capture additional information for the analysis of relationships between operands in $mov$ instructions. A more complete representation of the term distribution can be performed by repeating this process to construct dependency graphs for

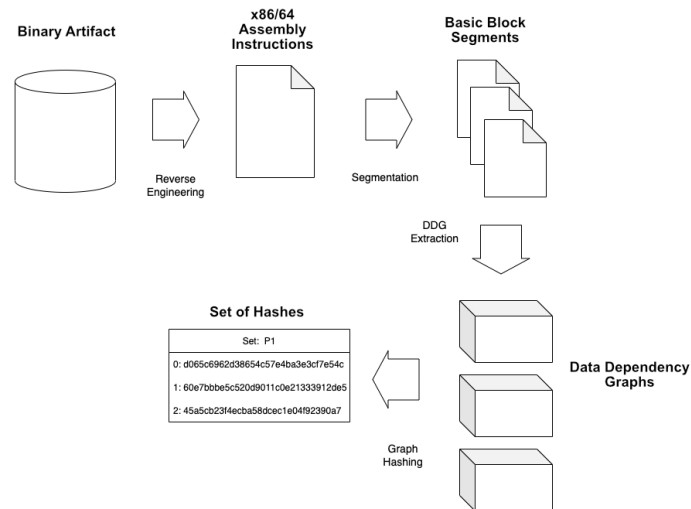

Figure 2: Feature extraction process for $DDG\ Fingerprinting$. Data dependency graphs represent patterns of data movement. The Weisfeiler-Lehman graph hashing algorithm is used to compare graphs for isomorphism, which are collected into a set, one per program.

every term. Further, we have represented dependencies using undirected graphs for simplicity. An additional resolution increase could be obtained by using directed graphs for each DDG graph.

### 2.2.3 GRAPH HASHING

Each data dependency graph was subsequently hashed for its isomorphic uniqueness using the Weisfeiler-Lehman graph hashing algorithm. This method yields a hash value which represents a single graph, such that two isomorphic graphs will correspond by having the same hash value. We use the $NetworkX$ library's implementation of the Weisfeiler-Lehman graph hashing algorithm for undirected graphs. Each basic block segment has a resulting hash which corresponds to its data dependency graph. Since we have segmented the program into basic block segments and have extracted data dependency graphs for each block, we can construct a set of hashes for each program binary sample Hagberg et al. (2008), Shervashidze et al. (2011).

Each program is then represented as a set of hashes, with each hash representing an isomorphically unique graph. The graphs are obtained from analysis of data dependency obtained after program segmentation, as discussed in the previous sections. This set represents a program's functionality as a collection of isomorphically unique patterns of data dependency. This representation enables a more detailed analysis of program features and their relationship to larger classes. We have represented each set (program) as a one-hot encoded vector. The dimensions of the vector space correspond to individual unique hashes identified for blocks of the programs in our dataset, and a vector's components are 0 or 1 to signify the absence or presence of a block with that hash code in the corresponding program Brownlee (2017).

Other hashing algorithms may also be considered, such as algorithms that produce semantically similar hashes. The selection of the hashing algorithm in our work was focused on representing graph isomorphism. The DDG Fingerprinting process is shown in Figure 2.

### 2.3 HAMMING SPACE

Next, we construct a metric space for our features based on Hamming Codes. Each vector is a program's Hamming Code and the distance metric between vectors is the Hamming Distance. The Hamming Code for a program is constructed by viewing the isomorphic hashes of the DDG Fingerprint as categorical values. Each isomorphic hash value is assigned one dimension in the vector

representation. Next, each each vector component is either 0 or 1, to represent the absence or presence in that program of a block segment with that isomorphic signature.

Formally, let $F = h_1, h_2, ..., h_m$ the set of all hash values of all blocks present in the library of program artifacts. An arbitrary ordering of elements is assumed during the construction of vectors. The Hamming Code for a program P is the vector $(v_1, v_2, ..., v_m)$, where $v_i$, $i = 1..m$ is 1 if the program P contains a block segment with an isomorphic signature $h_i$, and 0 if the isomorphic signature is not present.

The Hamming Distance is computed by computing the difference between two equal length strings of the one-hot encoded vectors Brownlee (2017), Hamming (1950), Tan et al. (2016).

As every isomorphic pattern in the library of examples contributes a dimension to the Hamming Code, the dimensionality of the space is very high. Our collection of examples has over 500 samples across multiple platform architectures, and the complete space has over 40k unique patterns of data dependency. Although the dimensionality is initially very large, the feature resolution can be adjusted once the specific characteristics of the search have been refined, which reduces the dimensionality to several hundred dimensions between a set of programs. We also use non-parametric methods for search, which are not as sensitive to high dimensional data, and are described in the next sub-section.

## 2.4 K-NEAREST NEIGHBORS

By creating a library of vectors with their Hamming Distances and composing a metric space, we can measure the similarity between vectors in terms of the distance metric. Vectors in our Hamming space with low distance correspond to semantically similar programs, due to: 1. DDG graphs reflect operationally semantically similar blocks; 2. the graph hashing process identifies and preserves DDGs similarity through isomorphism.

Therefore, when presented with a new program of an unknown class we extract the program's DDG Fingerprint, calculate the Hamming Code vector, and then query the set of examples. This can be done quickly and accurately by using k-Nearest Neighbors (k-NN). Since the distance metric in the space is Hamming Distance, we can retrieve the most semantically similar examples from the library of known programs for a new artifact with an unknown class. We present results for both the construction of the metric space using the Hamming Distance, and the k-NN search in the metric space in the following section Cover & Hart (1967), Hart et al. (2000).

## 3 EXPERIMENTAL RESULTS

### 3.1 QUANTIFYING OVERLAP OF FUNCTIONALITY

Our model makes it possible to answer a specific question: what is the degree of similarity that an unseen program has to an existing and previously seen program? Let us consider a malicious sample from the dataset, one file from the $ZeusGameover\_Feb2014$ Trojan malware binary.

To measure programs in terms of dissimilarity, a naive approach would compare across different operating systems, and so we can compare this malware sample to the GNU/Linux $ls$ program. It is likely that $ls$ will primarily read information from the filesystem. We expect a comparison of Trojan malware and the $ls$ program samples to not share many functional elements. We are able to quantify similarity of the operational semantics and perform further analysis. The total number of data dependency graphs collected is 234 for $ls$ and 622 for $ZeusGameover\_Feb2014$ sample 1. The set difference between the two sets will give us the degree to which the two programs are unique and differ from each other. The number of data dependency graphs that are present in $ls$ that are not present in the $ZeusGameover\_Feb2014$ sample is 121. The $ZeusGameover\_Feb2014$ sample set difference $ls$ has 509 unique data dependency graphs.

Another open question is what is the degree of functional overlap. We can measure the common functional patterns of data dependency between the two programs with the set intersection operation:

$$A \cap B$$

The intersection of $ls$ and $ZeusGameover\_Feb2014$ is 113.

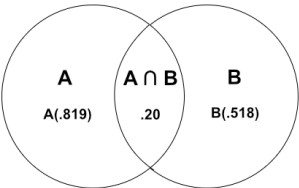

Figure 3: The overlap between malicious and benign samples is given by the Jaccard coefficient. This means that a median value of 20% of the structure of data movement within a program is shared between malicious and benign samples. However, in the median example, 80% of the functionality of malware is not shared with benign samples, and 50% of benign functionality is not shared in the malicious sample, or described by the set intersection.

| | Binary Name | DDG Count | Set Difference | Jaccard |
|---|---|---|---|---|
| 1 | $ZeusGameover\_Feb2014$, sample 1 | 622 | 509 | .179 |
| 2 | $ls$ | 234 | 121 | .179 |

Figure 4: Naive approach: a comparison between the Zeus Trojan and the GNU/Linux $ls$ program.

The degree of overlap between two sets can be determined by the Jaccard coefficient, which is the ratio of cardinalities of the set **intersection** and **union** Suppes (1972), Leskovec et al. (2020).

$$|A \cap B| / |A \cup B|$$

Since the set intersection is 113, we then calculate the union, which is 630. The Jaccard coefficient is then 113/630, or 0.179 Tan et al. (2016).

The degree of functional overlap is highly dependent on the selection of the samples in the library used to compose the Hamming space. However, one of the strengths of our program representation is that it offers the ability to adjust the level of feature resolution. Without increased resolution, it is difficult to interpret these results. For instance, we can repeat this process for the malicious Linux binary $Linux.Wirenet$ and benign samples collected from the $/usr/bin$ directory, which contains Unix system resources. For a comparison, we compare this malicious sample to a random set of benign Linux programs. These samples share at most a Jaccard coefficient of 0.270 with the malware. The minimum amount of overlap for these samples is 0.064, with a median of .204 Jaccard overlap. This means that the malicious binary typically shares 20% of its functionality with benign programs, and that these programs are also 80% dissimilar from the malware. Without an increase in feature resolution, it is challenging to know which properties are important to the class, the class composition, the degree to which the classes overlap, or the patterns that differentiate the class.

If we examine the benign Windows programs present in our dataset, then we can compare the $ZeusGameover\_Feb2014$ sample to the larger class of known functionality in system utilities. After deconstructing samples in the $System32$ directory to our benign dataset, we can then ask the question what is the largest degree of overlap between the benign class examples and a specific malware sample?

There is one and only one sample in the composed dataset that has a Jaccard coefficient of 1 with the Trojan malware. Based on the structure of data dependency we can discover that surprisingly $ZeusGameover\_Feb2014$ contains as a proper subset the system program $csrss.exe$. This utility is the $Client/ServerRuntimeSubsystem$ for Windows. $csrss.exe$ is also used as an exploitation mechanism for Trojan malware to corrupt a system. The method of exploitation used by the Trojan malware is not definitive to differentiate between classes, but the behavior has been discovered through search. The operational semantics have a degree of correlation identified by the feature representation. A fine grained analysis of the functionality enables further inferences regarding the correlation of functionality with a larger class. An unknown binary which includes benign code with known vulnerabilities as a proper subset is suspicious behavior, and could be used to

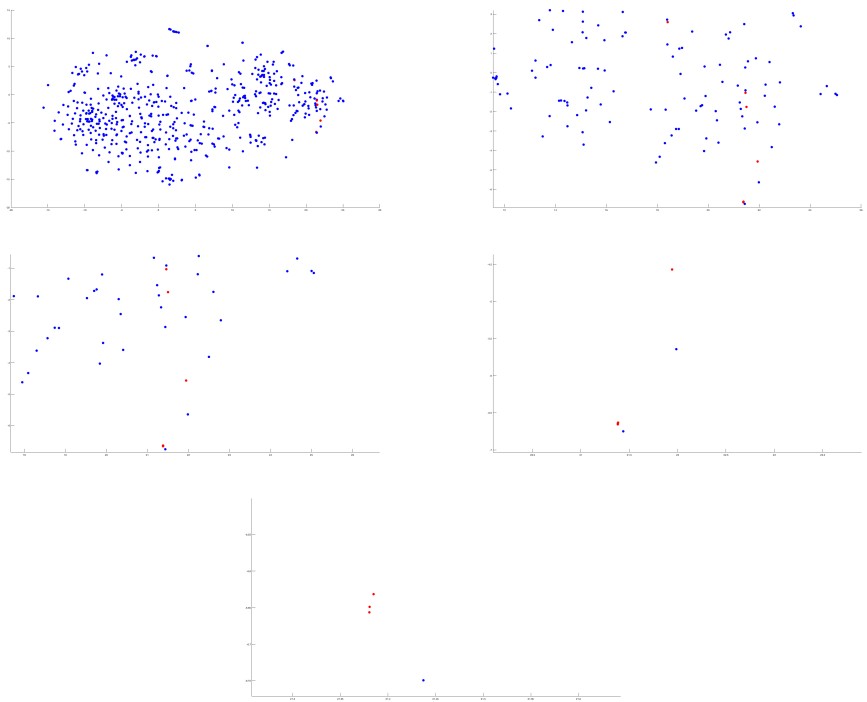

Figure 5: Progressive increases in resolution of the Hamming Space using t-SNE projection. This figure shows the $k$ neighbors identified in the high dimensional Hamming Space using kNN. Images of the space are presented in increased resolution in the appendix section. Malicious samples identified from search are highlighted in red.

disguise its functionality. A legitimate user would likely have privileges to access this specific system utility based on their level of access. Further exploration of the domain reveals known software vulnerabilities disclosed by CVE. This relationship between the programs was discovered from the analysis of the structure of their data dependency graphs, and by collecting hashes for each graph into sets National-Vulnerability-Database, Kallalike.

## 3.2 RESULTS OF K-NEAREST NEIGHBORS SEARCH

The examples in Figure 5 show a progressive increase in the resolution as the search is refined. This figure shows a projection of a high dimensional space using stochastic neighbor embedding ($t - SNE$). The search space is reduced as the resolution is increased progressively in each figure. This allows us to perform a more fine grained comparison of similarity between data points. We are able to determine boundaries between data points in the metric space based on their similarity. We are able to quantify the similarity between programs within a given space from the distance metric, and measure the degree to which programs are similar or different. This is useful in an adversarial environment when new program binary artifacts are provided without class labels or specifications. Quantitative analysis of similarity to examples is able to be acted on by an analyst or security policy, and this allows for classification results to be interpreted to a greater degree Hinton & Roweis (2002).

Search using k-NN returns a result of the $k$ closest data points in the dataset based upon the specified distance metric. In our metric space this is the set of data points with the lowest Hamming Distance from an example. A low Hamming distance is a measurement of a higher degree of overlap between data points and the selected example. Using this method we are able to answer questions related to the similarity and overlap of functionality between programs. The similarity of functionality was identified without a formal specification for verification. In an adversarial environment, a binary not previously seen with high similarity of functionality to a malicious program can be immediately identified and acted upon. The metric space allows for comparison of similarity based on the distance

| Neighbor | Hamming Dist. | % Difference | Program Name | Description |
|---|---|---|---|---|
| 1 | 91 | 0.21% | subst.exe | substitutes a virtual drive for a physical drive |
| 2 | 91 | 0.21% | dpapimig.exe | DPAPI Key Migration Wizard |
| 3 | 91 | 0.21% | TapiUnattend.exe | Telephony Unattend Action |
| 4 | 91 | 0.21% | wininit.exe | Windows Start-Up Application |
| 5 | 95 | 0.22% | fc.exe | DOS file compare utility |
| 6 | 95 | 0.22% | icsunattend.exe | no description available |
| 7 | 95 | 0.22% | regedt32.exe | Registry Editor Utility |

Figure 6: k-Nearest Neighbors search results with k=7 for $ZeusGameover\_Feb2014$. This table shows the indices for the 7 closest programs in a library of examples to an unseen malware example, along with the Hamming Distance from the malware to each neighbor. In order to find more fine-grained differences the resolution level can be increased based on these results, which are at the lowest level of resolution.

metric, and this is easily interpreted from the results of k-NN search. Further insights into specific datasets can be obtained from the measurement of similarity, and these are easily interpreted and visualized.

For the two malicious examples we have selected, we show the results of the k-NN search in Figures 6 and 7. We show the results for $k = 7$. The names of programs with the highest degree of similarity are listed along with the Hamming Distance from the selected malware example. This is useful to determine what functionality is present in the malicious sample. The functionality of the malicious sample is defined in terms of the known benign program functionality.

## 3.3 DISCUSSION

To our knowledge, data flow analysis has not been performed using features constructed in a bottom-up approach. This allows for an increase in resolution. This process yields very high dimensional vectors for a large search space, but the high dimensionality can be reduced by adjusting the feature resolution. A typical weakness of high dimensional spaces is the warping that occurs from the addition of higher dimensions. However, this is mitigated in the Hamming Space. An advantage to using k-NN is that an unseen example can be determined to be similar or different to a collection of existing examples that have been seen by the system. By creating a library of vectors with their Hamming Distances we can measure the similarity between vectors in terms of similarity via the distance metric. The similarity being measured is representative of features of data dependency graph isomorphism, and this is directly tied to both the structure and operational semantics of the program. We can quickly and accurately query the set of examples for a new example and receive the $k$ neighbors most associated with the vector based on the Hamming Distance. An advantage is that a new example can be determined to be similar or different to a collection of existing examples that have been seen by the system.

A disadvantage of this approach is that the Hamming vectors increase the dimensionality of the dataset, yielding high dimensional data, and require reduction. A strength of this approach is that it is computationally feasible, and that the similarity metric is an accurate representation of the program's functionality. This allows for an increase in interpretability. One weakness of this approach is that the Hamming Space must be recomputed based on the new data. When novel malware samples are encountered with behavior not previously seen, the Hamming Codes must be re-calculated. The cost of computation is the product of isomorphically unique hashes. But, this can be performed offline based on a specific period of time. Online learning was not the primary focus, but we intend to explore increases in efficiency and applications to real-time systems in future work.

A fundamental trade-off exists within our data between the level of resolution and the similarity. Low resolution is advantageous in quickly searching a large breadth in the search space. Once the search space has been narrowed at low-resolution and high dimensionality, a more fine-grained approach can be taken. As examples are analyzed with lower resolution they appear more similar and the distinguishing features are unclear. When the level of resolution is increased, differences are able to be discovered. Representing data with an adjustable resolution is advantageous for this reason. At high resolution levels, individual similarities and differences between samples can be shown clearly.

| Neighbor | H-Dist | % Difference | Program Name | Description |
|----------|--------|--------------|--------------|-------------|
| 1 | 338 | 0.78% | AtBroker.exe | Windows Assistive Technology Manager |
| 2 | 393 | 0.91% | wksprt.exe | RemoteApp and Desktop Connection Runtime |
| 3 | 393 | 0.91% | wowreg32.exe | SetupAPI 32-bit Surrogate |
| 4 | 397 | 0.92% | dllhost.exe | COM DLL library Hosting Surrogate |
| 5 | 406 | 0.94% | appidcertstorecheck.exe | AppID Certificate Store Verification Task |
| 6 | 406 | 0.94% | MRT-KB890830.exe | Malicious Software Removal Tool |
| 7 | 414 | 0.96% | cleanmgr.exe | Disk Space Cleanup Manager for Windows |

Figure 7: k-Nearest Neighbors search results with k=7 for $Win32.APT28.SekoiaRootkit$. This table shows the indices for the 7 closest programs in a library of examples to an unseen malware example, with the Hamming Distance from the malware to each neighbor. In order to find more fine-grained differences the resolution level can be increased based on these results, which are at the lowest level of resolution. We highlight that the $AtBroker$ executable is commonly used to disguise the behavior of malware.

Decision boundaries between examples can be determined as the resolution is increased. Quantitatively, the difference as measured by the percentage of DDG patterns that differ between samples increases as the resolution increases and dimensionality decreases. We use Jaccard coefficient to demonstrate the overlap between specific examples at high resolution levels.

In order to simulate an adversarial use case, a small set of unknown malicious binaries were selected and compared to a large class of benign examples. We have focused on benign examples of functionality for comparison, since obfuscation is a goal of an attacker. Additionally, since no specification exists for verification prior to execution in an adversarial environment, the binary file is the sole artifact available for analysis. While malicious programs at the level of binary files may have class labels, this is the lowest level of resolution, and often not descriptive of fine-grained program operation. We have shown that identification of malicious behavior and functionality on a fine-grained level, even when obfuscated, is possible using our representation as discussed in Section 3 and 3.1 of the Experimental Results and Figures 6, 7.

## 4    CONCLUSIONS

In this study we have proposed a new feature representation for binary programs, that is able to capture semantic and functional aspects of programs. We have collected a dataset of malicious and benign programs, and by segmenting them we extract graphs of data dependencies. We represented the isomorphic uniqueness of these graphs by hashing using Weisfeiler-Lehman graph hashing. Next, we collected data dependency hash values into a set of unique hashes for our artifact collection, and we constructed a search space using Hamming codes and the the Hamming distance. Using this new feature representation that we call DDG Fingerprinting, we are able to answer questions of overlap between specific executable instances and sets of programs to potentially analyze larger classes. We have performed search in the Hamming space with k-Nearest Neighbors.

This method is successful because data dependency is representative of operational semantics and structural properties of the program. Additionally, features were constructed in a bottom up approach, from which we are able to make additional inferences. This increase in accuracy and feature resolution allows programs to be compared in terms of their functionality, and this can be performed across platforms. In future studies we intend to explore the implications of increased resolution in semantic feature representations. Efficient search without prior training or deep learning represents an increase in accuracy, resolution, and interpretability.

### 4.1    REPRODUCIBILITY STATEMENT

The method and representation outlined in this study can be followed in future studies. We have outlined the process of data collection in Section 2.1 of the Methodology. This section describes the specific malicious examples that were selected, the location of the benign Windows system utilities, the tools used in the data collection process, and the method of feature extraction. The malware samples were chosen from a public repository of live malware, $theZoo$. The decompilation from binary

was performed using GNU *objdump*. Segmentation and the process of DDG Fingerprinting was performed using Python. The *NetworkX* library was used for processing graphs, and we used this library for graph hashing. Additionally we used *NetworkX* and Python to generate adjacency list representations of the undirected graphs used for data dependency. The results were obtained from analysis using Matlab and Python on the datasets outlined. Hamming Codes and Hamming Distance were calculated using the methods outlined in Section 2.3. Figures of the search space in Section 3.2 and the Appendix were generated using Matlab. Access to the dataset and implementation can be granted to reviewers upon request, and we intend to provide open access to these resources upon publication.

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

## A APPENDIX

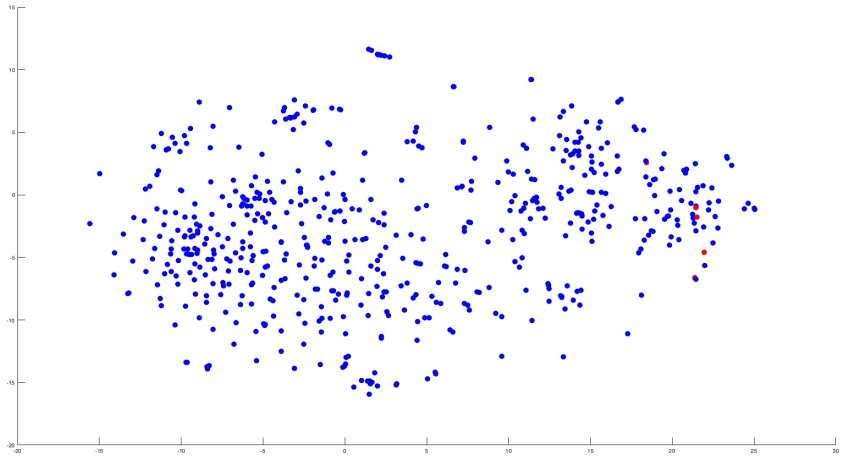

Figure 8: Resolution increase 1. t-SNE projection of Hamming Space for features of DDG Finger-prints for programs. Malicious samples identified from search are highlighted in red.

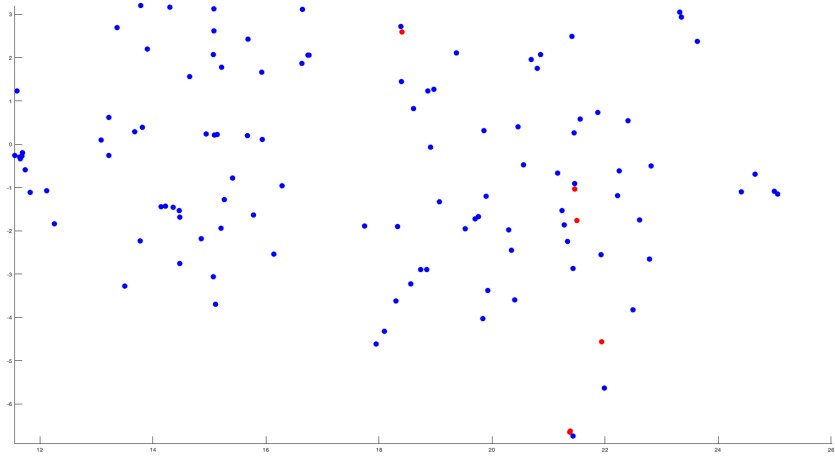

Figure 9: Resolution increase 2. t-SNE projection of Hamming Space for features of DDG Finger-prints for programs. Malicious samples identified from search are highlighted in red.

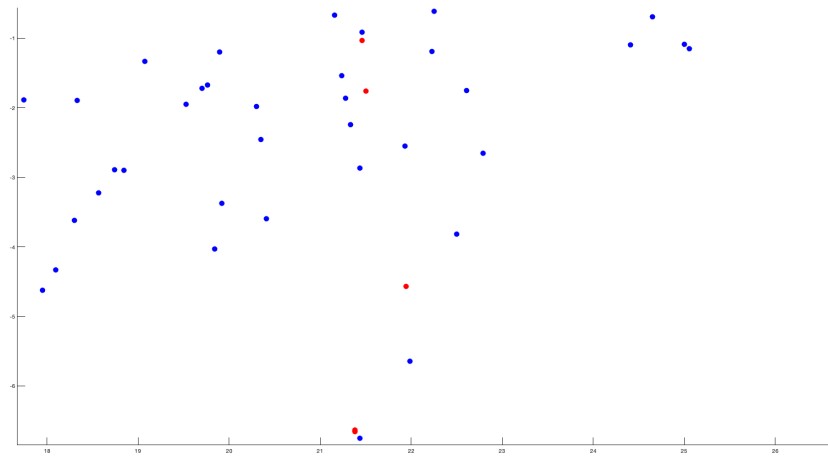

Figure 10: Resolution increase 3. t-SNE projection of Hamming Space for features of DDG Fingerprints for programs. Malicious samples identified from search are highlighted in red.

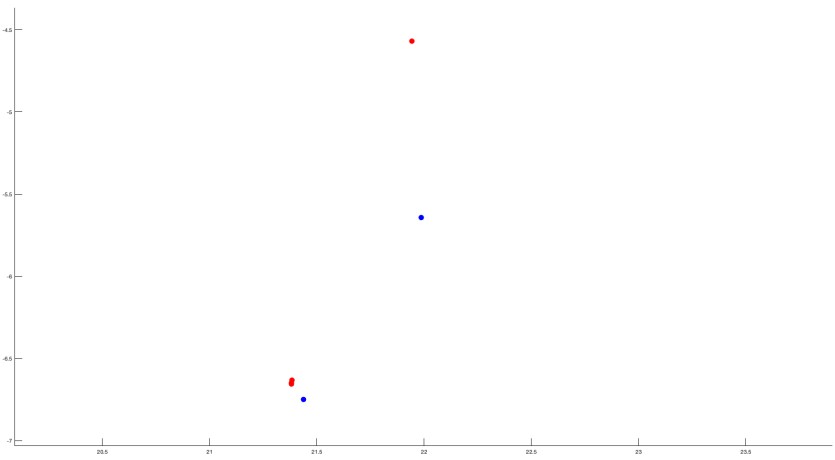

Figure 11: Resolution increase 4. t-SNE projection of Hamming Space for features of DDG Fingerprints for programs. Malicious samples identified from search are highlighted in red.

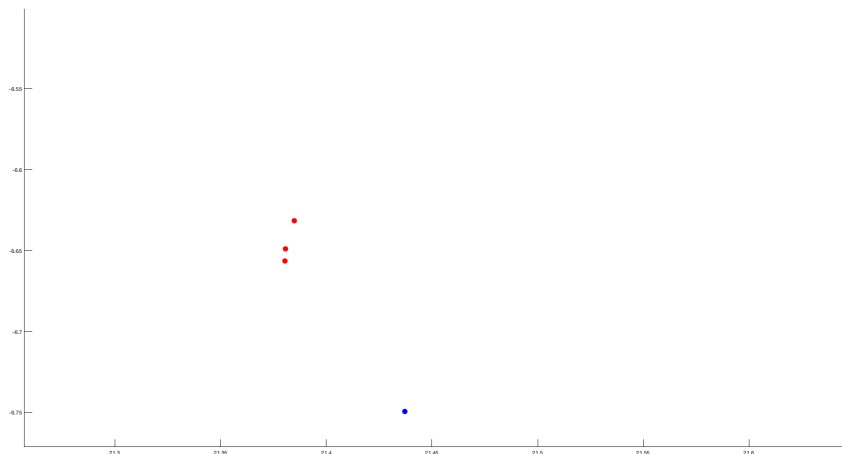

Figure 12: Resolution increase 5. t-SNE projection of Hamming Space for features of DDG Finger-prints for programs. Malicious samples identified from search are highlighted in red.

