# OpenReview forum: "Search and Retrieval in Semantic-Structural Representations of Novel Malware"
_ICLR.cc/2024/Conference — Submitted to ICLR 2024_

### Official Review · Reviewer_5Qou · 2023-10-19

**Soundness:** 2 fair
**Presentation:** 1 poor
**Contribution:** 2 fair
**Rating:** 3
**Confidence:** 4

**Summary:**

The authors propose a methodology for retrieving malware from a large corpus of data through a kNN algorithm applied on a novel feature extraction set, leveraging on Data Dependancy Graphs (DDG). Each program of the corpus is then expressed as a set of hashes that describe them, and that they can be used to be retrieved at need.
The authors describe some cherry-picked results to clarify how their methodology work, also showing an example of the first 7 neighbours of the Sekoia Rootkit.

**Strengths:**

1. Interesting retrieval approach that could be used to understand similarities between malware.
2. The approach is easy to understand.

**Weaknesses:**

**Missing a clear evaluation of the results.** The authors show one example retrieval, but all the discussion of Sect.3 is not enough to clarify the optimality and soundness of the proposal. In particular, how a practitioner could validate that the algorithm is picking up from the corpus of data meaningful neighbours? Also, the authors do no specify *why* the results of Fig.7 are relevant. What have these in common? The authors should provide some ground truth, trying to explain why they achieved those results.
Otherwise, if the methods would have retrieved other 7 samples, what would have been the conclusion?

**Possible errors in disassembly.** The authors state that they leverage *objdump* as disassembler. However, there are plenty of techniques that malware programs use to avoid reverse engineering. Usually, practitioners leverage other tools like IDA and Ghidra that are better in disassembly than objdump. Thus, the results might be biased towards the representation that is tool is providing, rather than capturing the indended functionality and graph shapes.

**Confused paper structure.** The manuscript would benefit for a re-arrangement of its structure. First, the abstract starts directly with the problem to solve, making it for newcomers to understand what is the problem to solve and why. Then, the introduction is missing which are the core contribution of the paper (hinted in Sect.2, but in a confused way). Most of the discussion is focused on just one cherry-picked example. There are no limitations, and no code is provided (that would have removed the need for Sect 4.1).

**Questions:**

1. Which are the advantage of using this method, and not other retrieval methods? The paper states the presence of related work, but none is compared to the proposed technique.
2. How this technique can be validated?

---

> ### Author Response · Authors · 2023-11-20
>
> Thank you for reviewing our work, and for your thoughtful comments and questions. Please see our responses below:
>
> We would be happy to re-order the manuscript based on structural changes.
>
> The question of reverse engineering refers to "techniques malware programs use to avoid reverse engineering".  Our discussion of the term reverse engineering refers to the process of translation from binary opcode to assembly during static analysis.  This is unrelated from other techniques that may be referred to as reverse engineering.  This is unrelated to dynamic analysis or obfuscation at runtime.  Objdump as a tool cannot perform control flow graph extraction, unlike Ghidra and IDA, so a comparison the recovered control flow graphs was not performed in this work, since control flow graphs were not used in the data set.  No intermediate representations were used, such a bytecode or LLVM IR.  As to the question of graph properties, we describe a method of data dependency graph extraction in detail, and our graphs were extracted from the direct static mapping of the binary.
>
> As to the evaluation of results, this method was only claimed to demonstrate the validity of the hypothesis, and better interpretability as a representation, not optimality of search performance.  A practitioner would be able to perform analysis on a more fine-grained level once specific functional overlap has been identified.  This overlap can be measured by the Jaccard coefficient as described in section 3.1 on quantifying functional overlap.  The conclusion presented is that functional overlap exists between the set of neighbors.  This is relevant because it provides a description of a malware sample as a collection of behaviors in relative terms.  This could be expanded to a comparison of other malware to find larger classes and their decision boundaries.  More specific details would require a fine-grained analysis of the behavioral properties, which we have shown through the measurement of similarity and the distance metric in the Hamming Space.  The correlation of the set of behaviors with the result can be measured by the Hamming Code of the sample.  If there is complete structural overlap with a behavior, we can begin to say that these are equivalent, a relation we intend to explore in future work.  Ground truth is verified through domain knowledge of the binaries included in the data set, so the results of search and retrieval will be accurate if the data has been collected, this is similar to a search engine indexing documents.  We have composed a library of benign examples and compared them to specific exemplars of malicious programs.

---

> > ### Comment · Reviewer_5Qou · 2023-11-20
> > **Thank you for the response**
> >
> > I personally thank the authors for replying to my comments.
> > However, I am still doubtful on basing all the data dependancy graph computation on the disassembler results given by objdump alone. Also, I think the paper needs to address clearly all the issues in a more systematic way.
> > For this reason, I am not changing my score.

---

> > > ### Author Response · Authors · 2023-11-20
> > >
> > > Thank you for your response.  To clarify, the data is not based on results reported from a single tool alone.  Disassembly is not a subjective interpretation, but defined strictly by the instruction set architecture.  This is the data present in the binary.  We are using a tool to perform the mapping from the binary to the instruction set definition of opcodes.  This is a 1-1 mapping, and there is no influence from the tool.  This is not the case for other types of reverse engineering, such as control flow graph recovery, where the graph properties do vary.  But, this analysis was not performed, and the data is not included in our study.

---

### Official Review · Reviewer_Qcjm · 2023-10-29

**Soundness:** 2 fair
**Presentation:** 2 fair
**Contribution:** 2 fair
**Rating:** 3
**Confidence:** 3

**Summary:**

The author proposes a novel representation method for binary programs. First, reverse engineering is employed to extract data dependency graphs (DDGs) from each program. Subsequently, a set of graph hashes is utilized to represent the distinct basic block segments within a DDG. By comparing the DDG Fingerprint of an unknown program with existing programs, the author employs k-Nearest Neighbors to determine its functionality. This approach enables the identification of the functionality of unknown programs through the comparison of DDGs.

**Strengths:**

The topic of searching and retrieving novel malware is both intriguing and significant.
The paper's structure and logic are lucid.
The discovery of a similarity between ZeusGameover Feb2014 and the Client/Server Runtime Subsystem is interesting.

**Weaknesses:**

The novelty of this method is limited, as it builds upon existing methods such as DDG, graph hash, and knn.
The experimental setup is overly simplistic, hindering the ability to effectively demonstrate the method's efficacy.

**Questions:**

1.The novelty of the proposed method is limited as it heavily relies on existing methods, thus the innovation of this paper is considered to be constrained.

2.The motivation behind selecting DDG as a feature is not adequately explained. Specifically, it is unclear what advantages this feature offers compared to other features such as control flow graphs or function call graphs in the context of software search and retrieval tasks.

3.The definition of Feature Resolution is excessively abstract, making it difficult to comprehend. It would be beneficial to provide an early explanation of this concept in the introduction section.

4.The DDG Fingerprint constructed by the authors appears to have a very high dimensionality, resulting in sparse data. Although the authors mention that “the feature resolution can be adjusted once the specific characteristics of the search have been refined”, I still struggle to understand how these specific characteristics are determined.

5.The experimental section is overly simplistic. 1), constructing a benign sample library with only 500 data is insufficient. 2), quantitative experiments are lacking. The authors only conducted qualitative analyses without providing accuracy metrics for identifying similarities between samples. 3), the use of only two selected samples in Figure 6 does not sufficiently establish the credibility of the results.

---

> ### Author Response · Authors · 2023-11-16
>
> Thank you for reviewing our work, and for your thoughtful comments and questions. Please see our responses below:
>
> Answer to Q1 (novelty of the proposed method): While the method makes use of existing algorithms, the method is used to demonstrate the hypothesis, which is that isomorphism of data dependency graphs is reflective of similarity of operational semantics, and the representation is also tied to structural properties. This is a novel claim, and validated by experimental results.
>
> Answer to Q2 (motivation behind using DDGs): There are several advantages of basing the representation of programs on data dependency graph features. They are directly representing operational semantics and structural properties of binary programs by definition. They can be observed at the lowest level in the architectural hierarchy, and constructed in a bottom up approach. The focus of this work is to measure semantic similarity with features tied to structural properties. We intend to demonstrate that this semantics persists across architectural layers in subsequent studies. This is not possible with function call graphs, which are not constructed in a bottom up approach but top-down.  Higher level language representations of functions may have many possible representations of a given binary. Prior work has studied isomorphism in control flow graphs, but this representation does not allow for increases in fine grained resolution, and all possible combinations of subgraphs must be enumerated for increased feature resolution. Control flow graphs can be used in labeled data sets, but the class labels are at the level of whole program, and this is the lowest level of detail. An efficient method to increase the feature resolution when using CFG-based representations has not been proposed yet.
>
> Answer to Q3: We are happy to re-organize the structure of the presentation. Resolution refers to the trade-off between scope of search at a coarse grained level, and accuracy of comparing samples at a fine grained level. Two programs that perform similar behaviors may appear to be similar when viewed at the level of all programs in the dataset, and based on all dimensions of the search space. However, they may have different goals, e.g. malicious vs benign – where the malicious sample obfuscates itself by using part of the benign code, and this could only be discovered upon closer inspection at a fine-grained level, and upon reduction of dimensionality of the search space, by removal of dimensions that do not contribute to differentiating the two samples. This is what was meant by “an increase in feature resolution.” However, this requires the search space to be reduced. This is analogous to image resolution, which is why the term was used.
>
> Answer to Q4: Refinement of search is based on a library of examples.  Based on examples of behavior to search for in a data set, the search space is refined by removing unnecessary examples from the search.  Similarity can be determined by the distance metric.  This can be used to answer the question, does the unknown program with an unknown class share any behavioral characteristics with known malware?  This can be used to answer the question of behavioral overlap, when programs are represented as a collection of behavioral or semantic characteristics. We have shown that the question of overlap can be answered for fine-grained analysis, and intend to explore the implications for larger classes of programs.
>
> Answer to Q5: Our dataset includes specific class exemplars of malicious programs, and compares them to a library of known benign programs. A single program contains thousands of basic block segments. While there were 500 benign programs in our library of behaviors, the total library of patterns of data dependency through DDG extraction was over 40,000 graphs, which were used for the relative comparison of a single class exemplar.
>
> Reporting the accuracy of a classifier on labeled data implies that the class label is correlated to a specific set of behaviors.  We have shown the degree of overlap between sets of behaviors, and that whether these are labeled as secure is a contextual question.  Accuracy requires increasing the level of feature resolution, and we have provided a representation to accomplish this task.  This question can be answered through the metric of similarity.
>
> The static reverse engineering process is a translation from the binary instruction which is mapped to the corresponding assembly instruction, no other transformation is required. This mapping is defined by the instruction set. There is no loss of data at this stage, it is a 1-1 translation.  This is a separate process from DDG graph extraction.

---

### Official Review · Reviewer_PPca · 2023-10-31

**Soundness:** 1 poor
**Presentation:** 1 poor
**Contribution:** 1 poor
**Rating:** 1
**Confidence:** 4

**Summary:**

The work presents an approach to generate malware representation using static analysis. That is, pieces of dependency graphs are built based on the assembly instructions of a given malware sample and then hashed using the Weisfeiler-Lehman graph hashing algorithm to retain the semantics of the dependency graphs and graph isomorphism. Finally, the hash values can be used to compute with Hamming Distance and KNN clustering. Only few evaluations with limited samples demonstrated the potential of the work.

**Strengths:**

- Clear presentation

**Weaknesses:**

It mainly lacks novelty and supportive arguments.
This work only adopts well-developed methods, including the Weisfeiler-Lehman graph hashing algorithm, Hamming Distance, and KNN, rather than developing a solution. Also, transforming static features of a given sample into hash values for further analysis is similar to some related work (Wu, et. al, 2021; Sun, et. al., 2022;), which also considers graph-based static features and maps to a representation for malware detection.

Wu, Chia-Yi, Tao Ban, Shin-Ming Cheng, Bo Sun, and Takeshi Takahashi. "IoT malware detection using function-call-graph embedding." In 2021 18th International Conference on Privacy, Security and Trust (PST), pp. 1-9. IEEE, 2021.

Sun, Qirui, Eldor Abdukhamidov, Tamer Abuhmed, and Mohammed Abuhamad. "Leveraging spectral representations of control flow graphs for efficient analysis of windows malware." In Proceedings of the 2022 ACM on Asia Conference on Computer and Communications Security, pp. 1240-1242. 2022.

**Questions:**

Questions:
- Is it possible to show the semantic preserving when assembly codes were transformed into blocks of dependency graphs and hash values?

Suggestion:
- A clear contribution can be shown with comprehensive evaluations of large-scale samples and compared with other approaches, such as control-flow malware variant detection (Cesare et. al., 2013).

---

### Official Review · Reviewer_pKB5 · 2023-11-03

**Soundness:** 2 fair
**Presentation:** 2 fair
**Contribution:** 2 fair
**Rating:** 3
**Confidence:** 4

**Summary:**

The author introduces a novel method for representing binary program features. This method utilizes data dependency to express the operational semantics and structural characteristics of the program, effectively capturing its semantic and functional aspects. Furthermore, the author introduces a bottom-up feature construction approach, enabling additional reasoning based on existing knowledge.

**Strengths:**

This article addresses a crucial field, considering the rapid proliferation of malware. Swift detection of zero-day malware and the identification of code reuse in zero-day malware present intriguing and formidable challenges.

The article introduces the DDG Fingerprinting method for detecting malware similarity, significantly enhancing the interpretability of detection outcomes.

**Weaknesses:**

In the Data Collection section, why were those two categories chosen, and other categories not considered? Are the malicious categories up-to-date?

Important terms should be further clarified, such as the frequently references ``resolution''. I couldn't find a clear and detailed definition or explanation of this term.

How are the issues encountered in reverse engineering addressed, such as code obfuscation and anti-debugging techniques? I believe reverse engineering is not a trivial matter, yet it is only briefly discussed.

In the Data Dependency Graph Extraction section, it isclaimed to be an undirected graph, yet Figure 1 shows a directed graph. Moreover, there is no explanation about 'ai' in Figure 1. Why is the 'mov' instruction singled out as capturing most changes without any data or experimental support to back this claim?

Figure 5 lacks detailed information on the horizontal and vertical axes, making it difficult to understand. More detailed analysis would be better.

**Questions:**

Please see the weakness section.

---

> ### Author Response · Authors · 2023-11-15
>
> Thank you for reviewing our work, and for your thoughtful comments and questions.  Please see our responses below:
>
> The goal of this study was to develop a description of program behavior in an adversarial environment, where behavioral descriptions are not available. This requires representing a program in relative terms to known behaviors and known programs, since neither the behavioral description or class are known. Programs that are not designed to be malicious may also have dangerous behaviors depending on context. The goal of this study was to analyze a program that was written with harmful intentions, and provide a description of the unknown behavior, and its similarity to known programs in relative terms. In this case two Trojan samples that had a large impact historically were selected as class exemplars to develop a description. So the samples under investigation were chosen as representatives of the class of Trojan malware to demonstrate the method, but this could be changed depending on search criteria and closer inspection of the behavioral description of a sample in the dataset.
>
> Regarding up-to-date malware samples, malware is changing rapidly, but these changes are at a syntactic level. The goal of this study was to provide a similarity of program semantics which is not sensitive to the underlying changes in syntax. So identifying semantic similarities in new malware is one of the primary goals. This was accomplished by providing a behavioral description of malware that has not previously been seen by the system. Expanding the dataset by adding malicious samples to the library of examples would increase the accuracy, and is not an impediment.
>
> Resolution refers to the trade-off between scope of search at a coarse grained level, and accuracy of comparing samples at a fine-grained level. Two samples that perform similar behaviors may appear to be similar when viewed at the level of all programs in the dataset, and based on all dimensions of the search space. However, they may have different goals, e.g. malicious vs benign, and this could only be discovered upon closer inspection at a fine-grained level, and upon reduction of dimensionality of the search space, by removal of dimensions that do not contribute to differentiating the two samples. This is what was meant by “an increase in feature resolution.” However, this requires the search space to be reduced.
>
> Code obfuscation is highly relevant to this work. We use similarity of graph isomorphism of data dependency to represent operational semantics. So the degree of obfuscation would be dependent upon the degree of semantic similarity preserved. We have selected to analyze a malicious sample which uses obfuscation to hide its behavior, and have shown that the behavior can be identified through program’s similarity to known samples.
>
> Anti-debugging techniques are used for dynamic analysis, and we have focused on static analysis. Dynamic analysis techniques would not prevent our method of static analysis to detect the behavior. This is because the pattern of data movement would be recognizable before execution at runtime in the static features. Although the values at runtime cannot be predicted, the structure of dependency allows for this analysis and comparison of similarity of behavior, and this was a primary goal of the study.
> The static reverse engineering process is a translation from the binary instruction which is mapped to the corresponding assembly instruction, no other transformation is required.
>
> Figure 1 is representing the extraction of a data dependency graph. The selection of a_i in Figure 1 is based on the set of operands in the instruction. Each instruction is composed of operands, and an edge between operands represents a dependency. We have shown a directed graph for the purposes of explanation because it illustrates the mapping of data dependencies in the example. Also, we included a short explanation of a_i and A_operand in Figure 1 caption.
> Undirected graphs were chosen in our dataset for the purposes of graph isomorphism, this was included as an area for increased accuracy in future work.
>
> The prevalence of the mov instruction was performed as part of previous studies. This can be shown easily through a histogram of term frequency of assembly instructions for a program.  We are happy to provide this figure as supplemental material for our data.
>
> Figure 5 is a projection of a high dimensional space constructed from the Hamming Distance. Each vector in the space represents a single sample and the associated Hamming Codes. This projection is used for visualization purposes, and quantitative analysis is used for accuracy. The axes shown in two dimensions are projections of a high dimensional space.  The metric space shown is to represent the distance metric in the constructed Hamming Space.  The distance metric is used to demonstrate similarity between samples.

---

### Meta-Review · Area_Chair_F92y · 2023-11-23

**Metareview:**

This work presents and evaluates a method for search and retrieval of malware from its binaries. This is achieved by using existing hashing methods over the data dependency graphs that preserve isomorphic uniqueness.

### Strengths
- This work aims to tackle an important problem.
- The method is conceptually simple and potentially robust.

### Weaknesses
- The evaluation does not compare and contrast this method with alternatives and baselines and does not fully reflect the complexity of the real task, as noted by the reviewers.
- The novelty wrt to machine learning methods is relatively small.
- The presentation of this work requires substantial more work.

Given the above, I recommend that the paper be rejected at its current state. Finally, I would argue that this work would be better evaluated in a security conference that is well-placed to understand the broad spectrum of the relevant literature and the needs of malware detection tool developers.

**Justification For Why Not Higher Score:**

Accepting this work won’t provide any valuable information to the ICLR community and will transfer the burden of a better evaluation from the authors to other researchers who want to advance the field.

**Justification For Why Not Lower Score:**

N/A

---

### Decision · Program_Chairs · 2024-01-16

Reject